# Triage of Critically Ill Patients: Characteristics and Outcomes of Patients Refused as Too Well for Intensive Care

**DOI:** 10.3390/jcm12175513

**Published:** 2023-08-25

**Authors:** Govind Sridharan, Yvan Fleury, Leila Hergafi, Sébastien Doll, Hatem Ksouri

**Affiliations:** Department of Intensive Care Medicine, Fribourg Hospital, CH-1700 Fribourg, Switzerland; yvan.fleury@h-fr.ch (Y.F.); leila.hergafi@h-fr.ch (L.H.); sdoll@citycable.ch (S.D.); hatem.ksouri@h-fr.ch (H.K.)

**Keywords:** intensive care, critical care, triage, admission, refusal, too well, delay, resource utilisation, outcome, mortality

## Abstract

Background: The appropriate selection of patients for the intensive care unit (ICU) is a concern in acute care settings. However, the description of patients deemed too well for the ICU has been rarely reported. Methods: We conducted a single-centre retrospective observational study of all patients either deemed “too well” for or admitted to the ICU during one year. Refused patients were screened for unexpected events within 7 days, defined as either ICU admission without another indication, or death without treatment limitations. Patients’ characteristics and organisational factors were analysed according to refusal status, outcome and delay in ICU admission. Results: Among 2219 enrolled patients, the refusal rate was 10.4%. Refusal was associated with diagnostic groups, treatment limitations, patients’ location on a ward, night time and ICU occupancy. Unexpected events occurred in 16 (6.9%) refused patients. A worse outcome was associated with time spent in hospital before refusal, patients’ location on a ward, SOFA score and physician’s expertise. Delayed ICU admissions were associated with ICU and hospital length of stay. Conclusions: ICU triage selected safely most patients who would have probably not benefited from the ICU. We identified individual and organisational factors associated with ICU refusal, subsequent ICU admission or death.

## 1. Introduction

As a costly part of a healthcare system, intensive care should be limited to patients who are expected to benefit and who cannot be treated on normal wards [1]. Admitting patients who do not need (i.e., are too well) or cannot benefit (i.e., too sick) from the intensive care unit (ICU) should be avoided even when there is no shortage of beds [2]. Although triage for ICU resulting in the refusal of some patients based on general and specific principles has been advocated for three decades [3,4], inappropriate patient selection leading to irresponsible ICU resource use still has been discussed recently [2]. The importance of ethical aspects and an equitable policy was heightened during the coronavirus pandemic [5,6], leading to the establishment of specific national guidelines in case of resource scarcity [7].

General recommendations for ICU triage have been proposed by different societies for intensive care [8,9,10,11]. This process takes into account an equitable distribution of available resources, potential benefits and harms for the patient, patient’s wishes and expected outcomes, with decisions about ICU care made at any stage of the hospital stay [12]. If ICU admission is requested, the intensivist has to state the patient’s eligibility, and the final decision should be in accordance with recommendations [13]. In real life, physicians tend to rely on their clinical judgment, patients’ comorbidities and functional status [14,15,16]. Other important determinants include management issues and patient assessment [17].

The final triage decision is either admission to the ICU or refusal for one of four distinct reasons: no bed available; patient does not wish; patient is too sick to benefit; patient is too well to benefit. Many studies have described the overall triage of patients not accepted to the ICU [18,19,20,21,22], and the difficult process of decision-making was highlighted [12,23]. Various factors influencing the triage decision in this population were described, such as individual patient parameters, hospital-related or other environmental factors, and physician-linked factors [24,25].

Studies focusing on patients deemed too well for the ICU are rare. We aimed to describe the characteristics of this specific population compared to patients admitted to the ICU, explore their outcomes, and analyse factors associated with unfavourable events after ICU refusal.

## 2. Materials and Methods

We conducted a retrospective single-centre, observational study at the Fribourg Hospital, Fribourg, Switzerland, a primary and secondary care centre with 400 acute care beds. The mixed 19-bed ICU was operated as a “close unit” with around 2000 admissions per year, including 12% of elective admissions. It was the only unit in the hospital for patients needing intermediate or intensive care.

All calls for ICU consultations recorded between 1 November 2016 and 31 October 2017 were screened for eligibility. Inclusion criteria were explicit requests for ICU admission with a final decision of either admission to the ICU or refusal for patients deemed too well. Exclusion criteria were requests coming from other hospitals; requests for consultation without explicit demand for admission; refusals due to bed unavailability; refusals for patients deemed too sick, including therapeutic limitations for the ICU; and patients for whom the intensivist advised transfer to a tertiary hospital. We handled successive consultations as followed: exclusion for subsequent refusal within 7 days; inclusion as new request for subsequent refusal after 7 days.

Requests for ICU admission took place on-call, without a rapid response team or electronic alert. For all critical situations, the specialist in charge, who could be either a resident or a senior physician, called the intensivist on duty. Each request was assessed by a senior intensivist or by an advanced fellow who could consult the senior at any time. There was no ICU triage protocol: the decision of admission or refusal was based on clinical judgement, taking into account all available data, including severity of illness based on current physiologic parameters, age, comorbidities and general health status, and patient’s or therapeutic representative’s preferences. The final decision was documented in the patient’s chart as admission to the ICU; no ICU bed available; refusal for low-risk patient (too well to benefit); or refusal if intensive care was considered inappropriate or not wished (too sick to benefit).

The data for each ICU request were collected in the electronic patient chart (refused patients) or extracted from the ICU registry (admitted patients), as shown in Appendix A. The Sequential Organ Failure Assessment (SOFA) score was based on all parameters available at time of request. Organ dysfunction was defined as a SOFA score of 2 or higher for each component.

Follow-up measures extracted from the patient’s chart included hospital LOS after triage decision and hospital mortality for all patients; occurrence of and reason for a delayed ICU admission within 7 days for previously refused patients; ICU LOS and mortality for all direct and delayed admissions.

All charts of patients initially refused were screened for an unfavourable outcome, defined as the occurrence within 7 days after refusal of either ICU admission without an obviously different reason or death without treatment limitations.

For the statistical analysis, all numerical variables were stratified into categories according to clinical relevance. The Chi-square test, or Fisher’s exact test as appropriate, was used to compare the baseline characteristics according to (a) refusal status (admitted versus deemed too well), and (b) occurrence of an unsuspected outcome. The same tests were used to compare LOS and mortality according to (c) refusal status, and (d) direct versus delayed ICU admissions. No assumptions were made for missing data.

Statistical analyses were performed using IBM SPSS Statistics 29.0 (IBM, Chicago, IL, USA). For all analyses, a *p* value less than 0.05 was considered significant.

The study was approved by the Swiss Ethics Committees on research involving humans (Project ID: 2017-01890). The application of the STROBE (Strengthening the Reporting of Observational Studies in Epidemiology) statement guidelines for observational cohort studies [26] is documented in Appendix A.

## 3. Results

During the 12-month study period, 2749 requests for ICU support were documented, and of these, 2488 were for ICU admissions (Figure 1). A total of 451 refusals were identified, corresponding to a global refusal rate of 18.1%. Among refused patients, 220 were excluded, with the two most frequent reasons being unavailability of ICU beds (3.9% of all requests), and patients deemed too sick for the ICU (2.5% of all requests). Among admitted patients, 49 (2.0% of all requests) were excluded as they were referred from another ICU.

A total of 2219 patients admitted to the ICU or deemed too well were included in the study, with a median age of 68.1 y (interquartile range (IQR) 55–78 y), and 38.8% were females. The most frequent diagnostic groups triggering requests for ICU admission were acute coronary syndrome (15%), other cardiovascular diseases (12%), respiratory disorders (14%), strokes (11%), and other neurological disorders (10%). Surgical patients were involved in 24.1% of requests (urgent surgery: 11.6%; elective surgery: 12.5%). Most requests came from the emergency department (58%), followed by the operating theatre or postoperative recovery room (24%), and the ward (15%).

Median time spent in the hospital before the ICU request was 5.8 h (IQR 2–26 h), with 82% of requests submitted within the first 2 days after admission to hospital. Treatment limitations had been applied in 15% of patients.

Most requests were received in the afternoon (41% from 12:00–18:00), and evening (31% from 18:00–24:00). Median ICU occupancy rate at time of request was 72% (IQR 67–83%).

### 3.1. Comparison of Included Patients According to ICU Triage Decision

Of the 2219 included patients, 1988 (89.6%) were directly admitted to the ICU, and 231 (10.4%) were deemed too well for the ICU. Among the latter, three patients had been refused a second time after the defined period of 7 days.

#### 3.1.1. Baseline Characteristics

The patients’ characteristics and organisational factors at ICU triage are shown in Table 1. There was no association between refusal status and age, sex or presence of a systemic oncologic disease. The triage decision was associated with the diagnostic groups, presence of treatment limitations, time already spent in hospital, surgical status, type of unit, day time and ICU occupancy.

#### 3.1.2. Outcomes

Hospital mortality for the entire study population was 7.5%. The triage decision was not associated with death in the ICU or in the hospital (Table 2). Among the 231 initially refused patients, an unfavourable outcome occurred in 16 (6.9%). Of these, two patients without any therapeutic limitations died 1–4 days after refusal, and 14 patients required ICU admission within 7 days.

### 3.2. Characteristics of Patients Deemed “Too Well” According to Event Occurrence

The 16 refused patients with an unexpected event had several baseline differences compared to the 215 patients with an eventless evolution (Table 3). A worse outcome was observed more frequently among patients with a LOS of more than 2 days before refusal. These patients also had higher SOFA scores at time of ICU request. The proportion of total SOFA scores and number of failing organs with at least 2 points was higher and affected the respiratory and haematologic components of the score (Appendix A). All events occurred after refusal by an experienced specialist.

### 3.3. Outcomes According to Delay in ICU Admission

Among the 2002 patients admitted to the ICU, 14 (0.7%) had been previously deemed too well. The majority of them were admitted to the ICU within 2 days after initial refusal. A delay in ICU admission was associated with longer ICU LOS and hospital LOS. ICU mortality was higher for delayed admissions, whereas the difference in hospital mortality was not significant (Table 4).

## 4. Discussion

The main finding of this study was that the ICU triage in our institution avoided one out of ten admissions by selecting patients who did not need the ICU. The process was safe, since 93% of patients deemed too well for the ICU had an eventless evolution.

The refusal rate for patients deemed too well for the ICU was 10.4%, while it varied widely in a 7–60% range in other studies [18,19,20,27,28,29,30,31,32]. Our low rate could be explained by a lower threshold of acceptance, and by the fact that our unit also received patients for intermediate care. One study reported, indeed, that the absence of a separate intermediate care unit lowered the refusal rate of patients deemed too well for the ICU [29].

ICU occupancy is one of the main external factors influencing refusal, as demonstrated by many studies [19,20,21,30,33,34]. One study found that units with fewer free beds refused more patients deemed as too sick, but not as too well [28]. In our study, the physicians tended to consider more patients as too well when fewer ICU beds were available, suggesting that they applied stricter criteria for ICU admission. This did not seem to affect outcomes, as the event rate was not associated with ICU occupancy.

Other external factors have been reported, accounting for the high variability of refusal rates between different institutions, even within the same study [19,31]. Our observed refusal rate was higher if the request came from the ward or if it was formulated during night time. We attributed this observation to the fact that our institution did not provide senior staffing on normal wards around the clock, generating more off-hour requests for ICU advice. An association between the refusal rate and the seniority of the referring physician has been described [27].

We observed that the refusal of patients deemed too well for the ICU was associated with individual patient factors, such as the principal diagnosis and surgical status, already reported by others [20,21,33]. Another factor that had not been described yet was the time already spent in hospital before the ICU request: for patients having already spent 2 days or more in hospital, the refusal rate tended to be higher. The fact that age was not associated with refusal could be explained by the low proportion of octogenarians who accounted for only 20% of our cohort. One study in the elderly found, indeed, that advanced age was a risk factor for refusal as too well [29].

Surprisingly, therapeutic limitations, mostly orders for “do not reanimate” (DNR), were associated with the refusal rate as too well in our study. DNR order was reported as an independent factor to ICU refusal in one study [35], but without the exclusion of patients deemed too sick. DNR orders should ideally not affect the physicians’ appreciation that a patient is too well for the ICU, but this factor may influence the clinical judgment.

Even though ICU underuse in women was described in large studies [36,37], ICU refusal was not associated with gender in our study. This is consistent with other studies [19,21,30,31,33], suggesting that the observed gender difference of ICU use is not due to the ICU triage itself, but to other factors.

Among patients deemed too well for the ICU, only 16 (6.9%) presented an unfavourable outcome. We chose an outcome combining 7-day mortality and delayed ICU admission within 7 days, in order to detect all patients who could have benefited from ICU admission. For the same reason, we chose the relatively long time span of 7 days while excluding two types of events that were not related to the initial refusal, i.e., death after therapeutic limitations or admissions for obvious other reasons, mainly post-operative monitoring. By this approach, we identified all triage decisions with a potentially harmful effect on the refused patients.

Overall hospital mortality for patients deemed too well for the ICU was 6.9% in our study, consistent with the mortality rates in the 6–12% range reported by others [18,19,27,30,31,38], whereas 2–3 fold higher rates were reported for octogenarians and cancer patients [29,32,33]. These relatively high mortality rates demonstrate that patients considered for the ICU but deemed too well have a different risk profile compared to the overall hospital population.

In our study, delayed ICU admission within 7 days without any obvious new reason occurred in 6.1% of the refused patients. Others reported a higher delayed admission rate of 9% despite taking into account only the 48 h following the refusal [39]. However, their refusal rate for patients deemed too well was 18%, almost double ours, likely explaining why more refused patients needed a secondary ICU admission.

Patients with delayed admissions had a longer LOS in the ICU and hospital, and three patients died after delayed admission, all of them in the ICU and more than 7 days after initial refusal, accounting for a hospital mortality of 21%. Mortality was not associated with delayed admission, a finding consistent with other studies of global population [18,21,39], whereas in cancer patients and patients needing mechanical ventilation, higher mortalities were reported after delayed ICU admission [32,40]. Some studies analysed patients accepted to the ICU but admitted with a variable delay due to waiting time, resulting in higher LOS in the ICU and in the hospital [41], and higher mortality [42,43]. This was also true for delayed ICU admission after the occurrence of physiological deterioration in medical patients [44].

Some factors were associated with unexpected outcomes among patients refused as too well for the ICU, such as the location on the ward at the time of the ICU demand and a LOS in hospital of more than 2 days prior to ICU request. However, these two factors were probably related to each other. Though patients whose ICU admission had been refused during the night could be more likely admitted later to an ICU [38], night-time triage was not related to the outcome in our study. Only two out of fifty patients refused between midnight and early morning were later admitted to the ICU, although our refusal rate was higher at this time slot.

Surprisingly, all patients with an unfavourable outcome in our cohort had been triaged by an experienced intensivist, although senior physicians should be the best trained for triage. This observation could be a coincidence, as only 44 (19%) patients were refused by fellows. Our hypothesis is that less experienced physicians admit any patient in case of doubt. As we did not record the grade of the physicians’ experience for direct admissions, we could not confirm this hypothesis, but others observed that physicians working regularly in the ICU [15] or having more experience [18] had a higher refusal rate.

We observed an association between SOFA score and unexpected outcomes. Unexpected events were indeed rare for patients having no more than one SOFA point, whereas almost two out of eight patients with at least 2 points presented an unfavourable outcome. Scores based on physiological parameters have already been shown to be useful for triage, for example, to identify patients who will survive if refused for the ICU [13]. Similarly, the modified early warning score is helpful to identify patients at risk of unplanned ICU admission after initial refusal [32,45]. As shown for patients deemed too sick, a decision tool could reduce inappropriate ICU admissions [46]. A model based on physiological parameters and diagnosis was developed to identify, among patients admitted to the ICU for monitoring, those who subsequently did not need active ICU treatment [47]. Even though it was not developed as a triage tool, such a model could help the intensivist better identify the patients who do not need the ICU. Whereas these scores or tools are not widely applied in clinical settings, the SOFA score is used in daily practice by many intensivists. We therefore suggest the integration of the SOFA score into the decision process of ICU triage.

It should be underlined that each score composed of physiological parameters has its limits. In our study, among the refused patients deemed too well, 37 (16%) had a SOFA score of 4 points or more, indicating at least moderate organ dysfunction, but only six of these patients had an unfavourable event. This observation illustrates that physicians should not rely on a scoring system alone, but combine it with their clinical judgement, as stated in recommendations for ICU triage [10].

Ultrasound was not used to guide triage decisions in our study. It has been described as a triage tool for specific diagnosis in the emergency department [48], or for pneumonia due to coronavirus disease in primary care [49]. Although some ultrasound-derived variables have been related to outcomes of patients admitted to the ICU [50], its role as an ICU triage tool beyond clinical assessment has not yet been defined.

Our study has several limitations. It was conducted at a single centre, rendering the generalisation to other settings difficult, as the triage process is highly dependent on the organisational structure. This was a retrospective observational study, as a prospective randomised design could not be justified on ethical grounds for studies on ICU triage. We did not collect information about functional status and frailty in refused or admitted patients. These parameters are an important issue for patients deemed too sick, but less for our study population. The follow-up for mortality was limited to LOS in the hospital; we considered longer time periods as unnecessary, as an unforeseen event of a patient deemed too well for the ICU would have likely occurred shortly after triage.

As the absolute number of unfavourable events was too small, we did not perform any further statistical analysis to identify independent risk factors for event occurrence, nor adjust them for confounding variables. The significant results in this subgroup should be interpreted as exploratory.

A selection bias is not excluded, as we included only patients who were formally referred for ICU admission. Other patients who were not presented for various reasons could have been missed.

## 5. Conclusions

The triage process for ICU admission in our institution identified most patients who could be treated on normal wards, avoiding the admission of patients deemed too well for the ICU. This process is safe, as demonstrated by an eventless evolution of the majority of these patients, despite a risk profile different from the general hospital population. We identified individual and organisational factors associated with ICU refusal as well as with subsequent ICU admission or death. Integrating the SOFA score into the process of ICU triage might be useful to further increase safety.

## Figures and Tables

**Figure 1 jcm-12-05513-f001:**
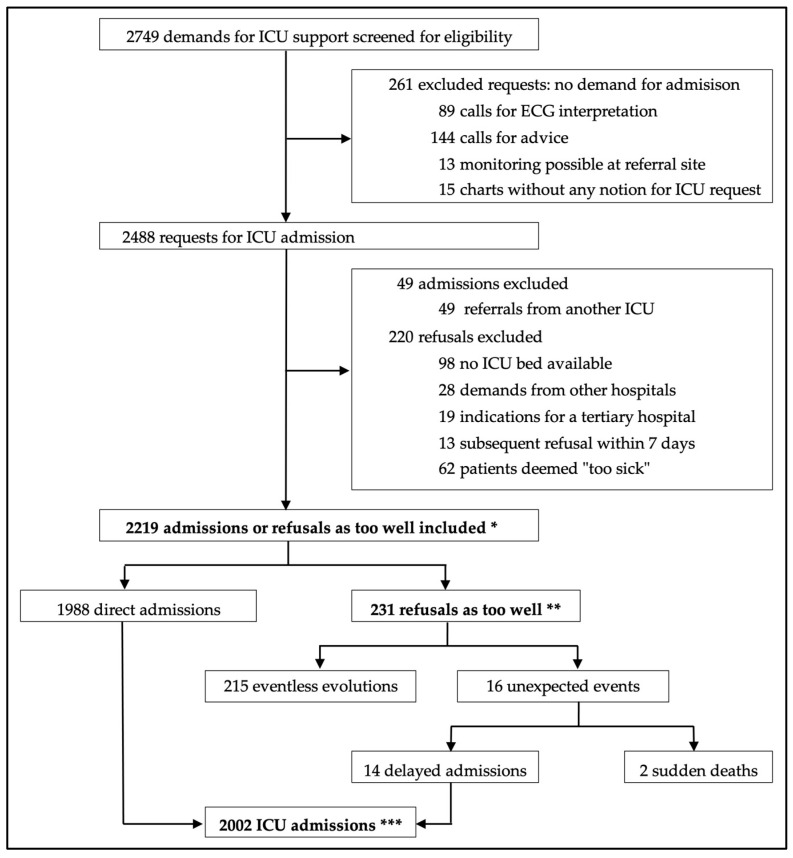
Flow chart of the screening process for requests for intensive care unit (ICU) consultation. Three analyses were performed for the groups shown in bold characters, to compare patients according to (1) * refusal status, (2) ** event occurrence, and (3) *** delay in ICU admission. ECG: electrocardiogram.

**Table 1 jcm-12-05513-t001:** Baseline characteristics stratified by ICU triage decision.

	Admitted to ICU	Refused as Too Well	*p* Value
	(N = 1988)	(N = 231)	
**Patient characteristics**			
Age distribution—no. (%)			0.82
14–64 y	840 (42.3)	102 (44.2)	
65–79 y	754 (37.9)	83 (35.9)	
80–98 y	394 (19.8)	46 (19.9)	
Sex–no. (%)			0.36
Male	1224 (61.6)	135 (58.4)	
Female	764 (38.4)	96 (41.6)	
Diagnosis for ICU request–no. (%)			<0.01
Acute coronary syndrome	313 (15.7)	27 (11.7)	
Arrhythmia	89 (4.5)	14 (6.1)	
Cardiovascular, other	230 (11.6)	42 (18.2)	
Respiratory, any	268 (13.5)	48 (20.8)	
Stroke	248 (12.4)	4 (1.7)	
Neurologic, other	189 (9.5)	23 (10.0)	
Sepsis	114 (5.7)	22 (9.5)	
Gastrointestinal, any	181 (9.1)	13 (5.6)	
Metabolic imbalance	77 (3.9)	20 (8.7)	
Intoxication	73 (3.7)	5 (2.1)	
Trauma	41 (2.1)	3 (1.3)	
Other reason ^1^	165 (8.3)	10 (4.3)	
Oncologic disease—no. (%)			0.51
None or limited disease	1786 (89.8)	209 (90.5)	
Metastatic cancer	126 (6.3)	18 (7.8)	
Leukaemia/Lymphoma	53 (2.7)	4 (1.7)	
Missing	23 (1.2)	0 (0.0)	
Limitations of life-sustaining treatments—no. (%)			0.02
No limitation	1711 (86.1)	181 (78.3)	
Any limitation ^2^	277 (13.9)	45 (19.5)	
Order “no ICU” ^3^	0 (0.0)	5 (2.2)	
Time since hospital admission—no. (%)			0.04
<2 d	1636 (82.3)	174 (75.3)	
2 to <7 d	174 (8.7)	28 (12.1)	
7 to 114 d	178 (9.0)	29 (12.6)	
**Organisational factors**			
Admission status—no. (%)			<0.01
Medical	1469 (73.9)	211 (91.3)	
Surgical, urgent	243 (12.2)	14 (6.1)	
Surgical, elective	272 (13.7)	6 (2.6)	
Missing	4 (0.2)	0 (0.0)	
Location at time of ICU demand—no. (%)			<0.01
Emergency room	1141 (57.4)	148 (64.1)	
Ward	273 (13.7)	68 (29.4)	
Postoperative recovery room	117 (5.9)	9 (3.9)	
Operating theatre	394 (19.8)	6 (2.6)	
Other location ^1^	63 (3.2)	0 (0.0)	
Time of demand for ICU admission—no. (%)			<0.01
00:00–05:59	263 (13.2)	50 (21.6)	
06:00–11:59	292 (14.7)	34 (14.7)	
12:00–17:59	820 (41.3)	81 (35.1)	
18:00–23:59	613 (30.8)	66 (28.6)	
Availability of ICU beds at time of ICU demand—no. (%)			<0.01
>4 beds	1046 (52.6)	97 (42.0)	
3–4 beds	573 (28.8)	73 (31.6)	
0–2 beds	369 (18.6)	61 (26.4)	

^1^ The categories “other reasons” and “other location” were not included in the analysis, as attribution was different between groups. ^2^ All therapeutic limitations were considered, except limitations for ICU admission. ^3^ Five patients deemed too well with a therapeutic limitation of “no ICU” were excluded from the specific analysis as not applicable to the admitted group.

**Table 2 jcm-12-05513-t002:** Outcomes according to ICU triage.

	Admitted to ICU	Refused as Too Well	*p* Value
	(N = 1988)	(N = 231)	
Death in ICU—no. (%) ^1^	90 (4.5)	7 (3.0)	0.29
Death in hospital, all—no. (%)	150 (7.5)	16 (6.9)	0.74
Death in hospital, according to treatment limitations at time of ICU demand—no./total of subgroup [%]			
No limitation	108/1711 [6.3]	9/181 [5.0]	0.48
Any limitation ^2^	42/277 [15.2]	6/45 [13.3]	0.75
Order “no ICU”	-	1/5 [20.0]	n.a.
Death within 7 days—no. (%)	85 (4.3)	4 (1.7)	0.06
No therapeutic limitations, no. (%) ^3^	-	**2 (0.9)**	n.a.
Therapeutic limitations, no. (%) ^4^	-	2 (0.9)	n.a.
ICU admission within 7 days, no. (%)	-	17 (7.4)	n.a.
No obviously new reason, no. (%) ^3^	-	**14 (6.1)**	n.a.
Other reason than at time of refusal ^5^	-	3 (1.3)	n.a.

^1^ For refused patients: if admitted later to the ICU during the same hospital stay (4 patients were admitted to the ICU more than 7 days after refusal). ^2^ Any therapeutic limitation was considered, except limitations for ICU admission. ^3^ Event defined as unfavourable outcome, the number of events are marked in bold characters. ^4^ Two patients died after a decision of palliative care at the ward. ^5^ Three patients were admitted to the ICU for post-operative monitoring that was not related to the initial refusal.

**Table 3 jcm-12-05513-t003:** Baseline characteristics of patients deemed too well stratified by event occurrence.

	Eventless Evolution	Unexpected Event ^1^	*p* Value
	(N = 215)	(N = 16)	
**Patient characteristics**			
Age distribution—no. (%)			0.16
14–64 y	96 (44.7)	6 (37.5)	
65–79 y	74 (34.4)	9 (56.2)	
80–98 y	45 (20.9)	1 (6.3)	
Sex—no. (%)			0.73
Male	125 (58.1)	10 (62.5)	
Female	90 (41.9)	6 (37.5)	
Diagnosis for ICU request—no. (%)			0.15
Acute coronary syndrome	26 (12.1)	1 (6.2)	
Arrhythmia	12 (5.6)	2 (12.5)	
Cardiovascular, other	42 (19.5)	0 (0.0)	
Respiratory, any	42 (19.5)	6 (37.5)	
Stroke	4 (1.9)	0 (0.0)	
Neurologic, other	23 (10.7)	0 (0.0)	
Sepsis	21 (9.8)	1 (6.3)	
Gastrointestinal, any	11 (5.1)	2 (12.5)	
Metabolic imbalance	18 (8.4)	2 (12.5)	
Intoxication	5 (2.3)	0 (0.0)	
Trauma	3 (1.4)	0 (0.0)	
Other reason ^2^	8 (3.7)	2 (12.5)	
Oncologic disease—no. (%)			0.53
None or limited disease	195 (90.7)	14 (87.5)	
Metastatic cancer	16 (7.4)	2 (12.5)	
Leukaemia/Lymphoma	4 (1.9)	0 (0.0)	
Limitations of life-sustaining treatments—no. (%)			1.00
No limitation	168 (78.1)	13 (81.2)	
Any limitation ^3^	42 (19.6)	3 (18.8)	
Order “no ICU”	5 (2.3)	0 (0.0)	
Time since hospital admission—no. (%)			0.04
<2 d	165 (76.7)	9 (56.2)	
2 to <7 d	23 (10.7)	5 (31.3)	
7 to 114 d	27 (12.6)	2 (12.5)	
SOFA Score, points			<0.01
0–1	121 (56.3)	2 (12.5)	
2–3	63 (29.3)	8 (50.0)	
4–5	26 (12.1)	4 (25.0)	
6–24	5 (2.3)	2 (12.5)	
**Organisational factors**			
Admission status—no. (%)			0.75
Medical	195 (90.7)	16 (100.0)	
Surgical, urgent	14 (6.5)	0 (0.0)	
Surgical, elective	6 (2.8)	0 (0.0)	
Location at time of ICU demand—no. (%)			0.04
Emergency room	142 (66.0)	6 (37.5)	
Ward	58 (27.0)	10 (62.5)	
Postoperative recovery room	9 (4.2)	0 (0.0)	
Operating theatre	6 (2.8)	0 (0.0)	
Time of demand for ICU admission—no. (%)			0.25
00:00–05:59	48 (22.3)	2 (12.5)	
06:00–11:59	29 (13.5)	5 (31.3)	
12:00–17:59	77 (35.8)	4 (25.0)	
18:00–23:59	61 (28.4)	5 (31.3)	
Availability of ICU beds at time of ICU demand—no. (%)			0.55
>4 beds	92 (42.8)	5 (31.2)	
3–4 beds	66 (30.7)	7 (43.8)	
0–2 beds	57 (26.5)	4 (25.0)	
Physician’s grade of experience—no. (%)			0.047
ICU specialist	171 (79.5)	16 (100.0)	
ICU fellow	44 (20.5)	0 (0.0)	
Written documentation of ICU refusal—no. (%)			0.40
Present	139 (64.7)	12 (75.0)	
Absent	76 (35.3)	4 (25.0)	
Duration of ICU consultation—no. (%)			0.78
1–15 min	132 (61.4)	10 (62.5)	
16–30 min	65 (30.2)	4 (25.0)	
31–90 min	18 (8.4)	2 (12.5)	
Consultation, type—no. (%)			0.71
At bedside	191 (88.8)	14 (87.5)	
By phone and chart	23 (10.7)	2 (12.5)	
By phone only	1 (0.5)	0 (0.0)	

^1^ An unexpected event was defined as either ICU admission without evidence of another indication, or death without treatment limitations for ICU, within 7 days after ICU refusal. ^2^ The diagnosis “other reasons” was not included in the analysis, as attribution was different between the two compared groups. ^3^ All therapeutic limitations were considered, except limitations for ICU admission.

**Table 4 jcm-12-05513-t004:** Outcomes for all patients admitted to the ICU, direct versus delayed admission.

	Direct Admission	Delayed Admission ^1^	*p* Value
	(N = 1988)	(N = 14)	
Deaths in ICU—no. (%)	90 (4.5)	3 (21.4)	0.02
Deaths in hospital—no. (%)	150 (7.5)	3 (21.4)	0.09
Deaths in hospital within 7 days—no. (%)	85 (4.3)	0 (0.0)	1.00
Death in hospital, according to treatment limitations at time of ICU demand—no./total of subgroup [%]			
No limitation	108/1711 [6.3]	2/11 [18.2]	0.13
Any limitation ^2^	42/277 [15.2]	1/3 [33.3]	0.24
Time between refusal and ICU admission—no. (%)			n.a.
<2 d	-	9 (64.3)	
2 to <4 d	-	4 (28.6)	
4 to 7 d	-	1 (7.1)	
ICU LOS—no. (%)			<0.01
<2 d	1442 (72.5)	7 (50.0)	
2 to <7 d	439 (22.1)	2 (14.3)	
7 to 60 d	107 (5.4)	5 (35.7)	
LOS in hospital after first ICU demand—no. (%)			0.02
<7 d	939 (47.2)	2 (14.3)	
7 to <14 d	542 (27.3)	5 (35.7)	
14 to 147 d	507 (25.5)	7 (50.0)	

^1^ Patients admitted to the ICU within 7 days after refusal, without an obvious other reason. ^2^ All therapeutic limitations were considered, except limitations for ICU admission. LOS: length of stay.

## Data Availability

The data that support the findings of this study are available from the corresponding author [G.S.], upon reasonable request.

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
