# Peer review of "Triage of Critically Ill Patients: Characteristics and Outcomes of Patients Refused as Too Well for Intensive Care"

_jcm, 2023, doi:10.3390/jcm12175513_

Round 1

Reviewer 1 Report

I read with interest the manuscript by Sridharan et al. on the characteristics and outcomes of patients refused as too well for intensive care. The article is sound and well written. I only have minor comments to be addressed:

- Line 35. I would also suggest to cite the COVID-19 pandemics, where the selection of patients for ICU admission became an ethical issue (doi: 10.1093/bmb/ldab009).

- Line 81-90. Please create a table with these information to increase readability.

- Line 122. Please replace "troubles" with "issues".

- Line 293. Besides the scores, the implementation of critical care ultrasound for clinical assessment of patients upon ICU admission has been proven to predict the possible prognosis of patients (doi: 10.1371/journal.pone.0182881). In fact, recent studies have shown that the incidence of cardiac dysfunction at ICU admission is often underestimated (doi: 10.1007/s00134-022-06685-2 - doi: 10.1007/s00134-023-07147-z) and associated with mortality (doi: 10.1111/echo.15462), thus suggesting the need of US assessment before ICU admission. Please discuss and add these 4 references.

Reviewer 2 Report

The authors present a retrospective single-center (medium-sized primary/secondary care center) study investigating the characteristics of patients denied ICU admission for being deemed too well compared to the patients accepted for ICU admission across a 1 year period. 2219 patients (1988, 90% ICU admits, 231, 10% ICU refusals). The analysis revealed that certain diagnoses for ICU request, medical rather than surgical admission, step up from ward, night time ICU request, and low ICU bed availability were the factors most associated with being denied ICU admission on the basis of being "too well". This is a well designed and conducted study on an important topic that has become exceedingly relevant currently as ICUs seem to be increasingly at capacity. The authors can be commended for uniquely investigating ICU triage to identify the factors that result in denial of an ICU bed. Most interestingly, institutional factors rather than patient factors contributed to ICU refusal, which supports the notion that ICU triage may be more subjective to institutional demands rather than clinical judgment on comorbidities and functional status.

Though time since hospital admission and SOFA score were identified as risk factors for eventual ICU admission, it would be worth comparing the clinical status of the 14 delayed ICU admissions at time of initial ICU refusal and at delayed admission to possibly reveal what threshold was reached to be deemed appropriate for ICU care.

Likewise, it would be valuable comparing the clinical status of the patients deemed too well for ICU during periods of sufficient and insufficient ICU bed capacity to possibly reveal what threshold was reached to be deemed appropriate for ICU care when resources were scarce.

Cardiovascular and Respiratory diagnoses were the greatest contributors to ICU request. At this reviewer's institution, certain respiratory interventions (ie Noninvasive mechanical ventilation and intubation) and certain medication given as infusions (vasopressors and antiarrhythmics) institutionally require ICU level care. Can the authors provide any breakdown on which proportion of patients admitted to the ICU vs refused as too well needed such interventions vs were though to require closer monitoring for impeding decompensation? 
